# Favorable prognosis of patients who received adjuvant androgen deprivation therapy after radiotherapy achieving undetectable levels of prostate-specific antigen in high- or very high-risk prostate cancer

Jae-Uk Jeong[1], Taek-Keun Nam[1]*, Ju-Young Song[1], Mee Sun Yoon[1], Sung-Ja Ahn[1], Woong-Ki Chung[1], Ick Joon Cho[1], Yong-Hyub Kim[1], Shin Haeng Cho[1], Seung Il Jung[2], Taek Won Kang[2], Dong Deuk Kwon[2]

1 Department of Radiation Oncology, Chonnam National University Hwasun Hospital, Chonnam National University College of Medicine, Hwasun, Korea, 2 Department of Urology, Chonnam National University Hwasun Hospital, Chonnam National University College of Medicine, Hwasun, Korea

* tknam@jnu.ac.kr

**Data Availability Statement:** Data cannot be shared publicly because of potentially identifying or

## Abstract

### Introduction

To determine the prognostic significance of long-term adjuvant androgen deprivation therapy (A-ADT) over 1 year in achieving undetectable levels of prostate-specific antigen (PSA) less than 0.001 ng/mL in prostate cancer patients with high- or very high-risk prostate cancer who underwent radiotherapy (RT).

### Materials and methods

A total of 197 patients with prostate cancer received RT, with a follow-up of $\geq$12 months. Biochemical failure was defined as PSA $\geq$nadir + 2 ng/mL after RT. We analyzed clinical outcomes, including survival, failure patterns, and prognostic factors affecting outcomes.

### Results

Biochemical failure-free survival (BCFFS), clinical failure-free survival, distant metastasis-free survival, cancer-specific survival, and overall survival (OS) rates at 5 years were 91.1%, 95.4%, 96.9%, 99.5%, and 89.1%, respectively. Administration of long-term A-ADT significantly predicted favorable BCFFS ($p = 0.027$) and OS ($p < 0.001$) in multivariate analysis. Nadir PSA $\leq$0.001 ng/mL was an independent prognostic factor for BCFFS ($p = 0.006$) and OS ($p = 0.021$). The use of long-term A-ADT significantly affected nadir PSA $\leq$0.001 ng/mL ($p < 0.001$). The patients with A-ADT for 1 year or longer had better BCFFS or OS than those for less than 1 year or those without A-ADT ($p < 0.001$). The best prognosis was demonstrated in patients treated with long-term A-ADT and nadir PSA $\leq$0.001 ng/mL in BCFFS ($p < 0.001$).

sensitive patient information. Data are available from our Institutional Review Board for researchers who meet the criteria for access to confidential data: cuhhirb@gmail.com.

**Funding:** This study was financially supported by Chonnam National University (Grant number: 2017-0227).

**Competing interests:** The authors declare that there is no conflict of interest regarding the publication of this paper.

## Conclusion

The addition of long-term A-ADT over 1 year to RT demonstrated good treatment outcomes in patients with locally advanced prostate cancer. Achieving a nadir PSA value ≤0.001 ng/mL using combination therapy with RT and A-ADT is a powerful clinical predictor of treatment outcomes.

## Introduction

The prostate is the third most common site of cancer incidence in Korean men older than 64 years in 2016 [1]. As the incidence of prostate cancer increases, there is a trend that the proportion of locally advanced prostate cancer also increases [2]. Those who are considered to have high risk are defined by the National Comprehensive Care Network (NCCN) as having at least one of the following features: T3a, Gleason Group 4 or 5, and serum prostate-specific antigen (PSA) value of more than 20 ng/mL [3]. Those with at least one of the following features, T3b-4, primary Gleason pattern 5, or >4 cores with Gleason Group 4 or 5, are defined as being at very high risk. Radiation therapy (RT) for these patients could be a therapeutic option among various modalities [4].

Currently, RT with androgen deprivation therapy (ADT) is the standard treatment of choice for high-risk patients. Neoadjuvant ADT (N-ADT) and concurrent ADT (C-ADT) could improve the prognosis of intermediate- and high-risk patients [5, 6]. Several randomized trials have illustrated the efficacy of adjuvant ADT (A-ADT) administered with RT in locally advanced prostate cancer [7–9]. Additionally, long-term A-ADT could improve treatment outcomes in high-risk patients treated with RT [4]. A randomized trials demonstrated that A-ADT for 34 months resulted in better treatment results than A-ADT for 4 months [7]. However, the optimal duration of A-ADT is unclear in the setting of RT.

We investigated treatment outcomes and related predictive factors in a single institute cohort of patients with high- or very high-risk prostate cancer without regional lymph nodal involvement who received RT. The aim of this study was to evaluate the efficacy of long-term A-ADT and to determine other prognostic factors associated with patient- or treatment-related characteristics in the setting of RT.

## Materials and methods

### Patients

Radiotherapy was administered to 306 patients with prostate cancer between January 2005 and December 2015 at Chonnam National University Hospital. Among these patients, we excluded 42 patients with low or intermediate risk, 35 patients with double primary cancer or distant metastasis at diagnosis, 26 patients with regional lymph node metastasis, three patients treated using three-dimensional conformal RT (3D-CRT) only, three patients with incomplete medical records after RT. A total of 197 patients with high or very high risk according to NCCN risk group [3] were included in this retrospective analysis. During the follow-up period, regular serum PSA level measurements were performed every 3 months for the first 5 years and every 6 months thereafter. Imaging workups, including computed tomography, magnetic resonance imaging, whole-body bone scintigraphy, and positron emission tomography/computed tomography, were performed in patients under suspicion of recurrence. Informed consent waiver was permitted according to the federal regulation of Department of Health and Human

Services (code title 45, 46.116) and this retrospective study was approved by the Institutional Review Board of Chonnam National University Hwasun Hospital (CNUHH-2019-099).

## Treatments

The RT techniques used in patients included intensity-modulated RT (IMRT) or a combination of 3D-CRT and IMRT without brachytherapy. For bladder and rectal preparations, each patient was instructed to empty the bladder and rectum, and then drink two cups of water 1 hour before the acquisition of planning CT and each treatment. To verify the patient's position, daily electronic portal images and weekly cone beam CT images were obtained using a linear accelerator, or daily megavoltage CT was obtained using helical Tomotherapy. Generally, RT was administered equal or greater than 70-Gy equivalent dose in 2-Gy fractions with α/β ratio of 2.0 ($EQD_{2/2}$) using 2.0–2.2-Gy fraction size. RT was delivered using two consecutive plans. The first plan included the prostate and whole seminal vesicle or pelvic lymphatics if necessary. The second plan included the prostate and lower half of the seminal vesicle, or whole seminal vesicle if invaded. Patients with a probability of pelvic lymph node involvement of greater than 20% according to the Roach formula, $\frac{2}{3} \times PSA + (Gleason\ score - 6) \times 10$, were candidates for whole pelvic RT, including the internal iliac, external iliac, and common iliac lymph nodes [10]. Generally, ADT consisted of luteinizing hormone-releasing hormone analogue (LHRH) agonist with or without an antiandrogen. The durations of neoadjuvant ADT (N-ADT) and adjuvant ADT (A-ADT) were defined as $\geq 2$ months before RT and $\geq 1$ year after RT, respectively. Concurrent ADT (C-ADT) was performed during RT.

## Statistical analysis

Biochemical failure (BCF) was defined as serum PSA level $\geq$nadir + 2.0 ng/mL [11]. Biochemical failure-free survival (BCFFS) was defined as the time between the first date of RT and the date of BCF. Clinical failure-free survival (CFFS) was defined as the time between the first date of RT and the date of disease progression, including locoregional failure and distant metastasis on imaging or histology. Distant metastasis-free survival (DMFS) was defined as the time between the first date of RT and the date of metastases to distant organs or non-regional lymph nodes on imaging or histology. Cancer-specific survival (CSS) was defined as the interval between the first date of RT and the date of death from progressive prostate cancer. Overall survival (OS) was defined as the duration between the first date of RT and the date of the patient's death or censorship on the final date of follow-up.

Variables analyzed for predicting outcomes were as follows: age at RT start ($\leq$70 vs. >70 years), initial PSA ($\leq$20 vs. >20 ng/mL), clinical T stage (<3b vs. $\geq$3b), Gleason grade group (<5 vs. $\geq$5) [12], NCCN risk group (high vs. very high), N-ADT before RT (no vs. yes), A-ADT after RT (no vs. yes), RT volume (prostate vs. whole pelvis), $EQD_{2/2}$ ($\leq$76.23 vs. >76.23 Gy), and nadir PSA after RT ($\leq$0.001 vs. >0.001 ng/mL). All 10 variables were used in univariate and also in multivariate analysis.

Radiation-related toxicity was evaluated according to the Common Terminology Criteria for Adverse Events (version 5.0). Kaplan–Meier models were used for survival analysis of all potential factors that affected treatment results and were tested using the log-rank test. A Cox proportional hazards model was used for multivariate analysis. A chi-squared test was used to evaluate the relationship between two categorical variables. On statistical analysis, $p$-values less than 0.05 were considered significant. All statistical analyses were performed using SPSS version 19.0 (IBM SPSS, Armonk, NY, USA).

## Results

### Patients and treatment characteristics

A total of 197 patients with high (n = 86) or very high risk (n = 111) were included in this study (Table 1). The median age was 71 years (range, 52–83 years) and the median initial PSA was 20.21 ng/mL (range, 1.18–253). Almost all patients had an advanced T stage of 3 or 4 (n = 144). The most common Gleason grade group was 4 (n = 60), followed by 5 (n = 42). All patients received ADT except for four patients due to the history of cardiovascular event. Considering the type of ADT, N-ADT was delivered to 78 patients (39.6%), C-ADT was administered to 188 patients (95.4%), and A-ADT was administered to 160 patients (81.2%). The total duration of A-ADT ranged from 12.4 to 94.3 months with a median 36 months. The radiation field was confined to the prostate in 26 patients and to the whole pelvis in 171 patients. The nominal dose of RT ranged from 66.0 to 77.7 Gy. The median nominal dose of radiation and $EQD_{2/2}$ were 72.6 Gy and 76.23 Gy, respectively. The dose fractionation schedule mostly consisted of 2.2 Gy fractions (n = 164). The median nadir PSA level after RT was 0.001 ng/mL.

### Treatment outcomes and pattern of failure

The median follow-up duration was 72 months (range, 14–96 months). Overall, the estimated 5-year BCFFS, CFFS, DMFS, CSS, and OS rates were 91.1%, 95.4%, 96.9%, 99.5%, and 89.1%, respectively (Fig 1). During the follow-up period, BCF was seen in 18 patients (9.1%). Clinical failure was observed in 10 patients (5.1%): in three patients with loco-regional failure and in seven patients with distant metastasis. Sites of distant metastasis were located in the bone in five patients and in the lungs in two. Two patients (1.0%) died due to progression of prostate cancer among the 23 patients (11.6%) who died during the follow-up period.

### Prognostic factors related to treatment outcomes

Age ($p = 0.001$), initial PSA ($p = 0.014$), T stage ($p = 0.050$), A-ADT ($p < 0.001$), and nadir PSA ($p < 0.001$) were significant prognostic factors for BCFFS in univariate analysis. Age ($p = 0.012$), T stage ($p = 0.017$), NCCN risk group ($p = 0.049$), and nadir PSA ($p = 0.008$) were significantly associated with CFFS. T stage ($p = 0.010$) and NCCN risk group ($p = 0.026$) were significant prognostic factors for DMFS. A-ADT ($p < 0.001$) and nadir PSA ($p = 0.004$) significantly affected OS (Table 2 and Fig 2). The results of the multivariate analysis of prognostic factors are shown in Table 3. Age ≤70 years ($p = 0.005$) and initial PSA level >20 ng/mL ($p = 0.044$) were poor prognostic factors for BCFFS. Administration of A-ADT was a good prognostic factor for BCFFS ($p = 0.027$) and OS ($p < 0.001$). Nadir PSA was a significant prognostic factor for BCFFS ($p = 0.006$) and CFFS ($p = 0.021$). Of the 37 patients who did not receive A-ADT, only 7 patients (18.9%) achieved nadir PSA ≤0.001 ng/mL. However, 92 patients (57.5%) of 160 patients who administered A-ADT reached nadir PSA ≤0.001 ng/mL ($p < 0.001$).

The duration of A-ADT significantly affected BCFFS ($p < 0.001$; Fig 3). The 5-year BCFFS rates were 74.9% in patients without A-ADT, 84.0% in patients received A-ADT with less than 2 years, and 96.6% in patients received A-ADT with equal or more than 2 years. Patients were stratified into four subgroups based on A-ADT and nadir PSA: nadir PSA ≤0.001 ng/mL with or without A-ADT, nadir PSA >0.001 ng/mL with or without A-ADT. A-ADT and nadir PSA reached ≤0.001 ng/mL was associated with the best prognosis for BCFFS among the three groups ($p < 0.001$; Fig 4).

**Table 1. Patients and treatment characteristics.**

| Characteristics | | Number (%) |
|---|---|---|
| Age | Median | 71 |
| | Range | 52–83 |
| Initial PSA level (ng/mL) | Median | 20.21 |
| | Range | 1.18–253.00 |
| Clinical T stage | 1c | 5 (2.5) |
| | 2a | 12 (6.1) |
| | 2b | 8 (4.1) |
| | 2c | 28 (14.2) |
| | 3a | 50 (25.4) |
| | 3b | 88 (44.7) |
| | 4 | 6 (3.0) |
| Gleason Group | 1 | 24 (12.2) |
| | 2 | 38 (19.3) |
| | 3 | 30 (15.2) |
| | 4 | 60 (30.5) |
| | 5 | 42 (21.3) |
| | Unknown | 3 (1.5) |
| NCCN risk group | High | 86 (43.7) |
| | Very high | 111 (56.3) |
| ECOG | 0 | 114 (57.9) |
| | 1 | 83 (42.1) |
| Type of ADT | LHRH agonist + antiandrogen | 133 (67.5) |
| | LHRH agonist | 60 (30.5) |
| | No | 4 (2.0) |
| N-ADT | No | 119 (60.4) |
| | Yes | 78 (39.6) |
| C-ADT | No | 9 (4.6) |
| | Yes | 188 (95.4) |
| A-ADT | No | 37 (18.8) |
| | Yes | 160 (81.2) |
| Duration of A-ADT (months) | Median | 36 |
| | Range | 12.4–94.3 |
| RT dose (Gy) | Median | 72.6 |
| | Range | 66.0–77.7 |
| RT dose, $EQD_{2/2}$ | Median | 76.23 |
| | Range | 70.00–80.85 |
| Fraction size (Gy) | 2.0–2.1 | 25 (12.7) |
| | 2.2 | 164 (83.2) |
| | Mixed | 8 (4.1) |
| RT volume | Prostate only | 26 (13.2) |
| | Whole pelvis | 171 (86.8) |
| RT mode | 3D–CRT and IMRT | 9 (4.6) |
| | IMRT | 188 (95.4) |
| Nadir PSA after RT (ng/mL) | Median | 0.001 |
| | Range | 0.000–2.934 |
| Duration of follow-up period (months) | Median | 72 |

(*Continued*)

**Table 1.** (Continued)

| Characteristics | | Number (%) |
| --- | --- | --- |
| | Range | 14–96 |

PSA, prostate-specific antigen; NCCN, National Comprehensive Care Network; ECOG, Eastern Cooperative Oncology Group; LHRH, luteinizing hormone-releasing hormone analogue; N-ADT, neoadjuvant androgen deprivation therapy before radiotherapy; C-ADT, concurrent androgen deprivation therapy with radiotherapy; A-ADT, adjuvant androgen deprivation therapy after radiotherapy; RT, radiotherapy; Gy, gray; $EQD_{2/2}$, equivalent dose in 2 Gy fractions at α/β ratio of 2.0; 3D-CRT, three-dimensional conformal radiotherapy; IMRT, intensity-modulated radiotherapy.

## Treatment-related toxicity

Treatment-related toxicity was observed within acceptable levels during the follow-up period (Table 4). Regarding acute genitourinary complications, patients complained urinary frequency (44.7%) and dysuria (12.7%), frequently. Diarrhea and anal pain were reported in 39 patients (19.8%) and 24 patients (12.2%), respectively. There were no grade 3 or 4 acute toxicities. Late grade 3 genitourinary toxicities were observed in two patients, one patient with gross hematuria requiring transfusion and the other with acute urinary retention requiring intervention. Three patients (1.5%) received argon plasma laser coagulation for late grade 3 hemorrhagic proctitis. There were no grade 4 late toxicities. Although we could not conclude the causal relationship between ADT and related complications, some patients developed complications, which might have been correlated with administration of ADT as follows. Hot flashes, which were self-limited, developed in 17 patients (8.6%). Three patients (1.5%), who received ADT over 36 months, experienced thromboembolism in lower legs and received thrombectomy. Myocardial infarctions occurred in three patients (1.5%), who received ADT for 3, 6, and 12 months, respectively, and received percutaneous coronary intervention. One patient, who received ADT for 47 months, developed cerebral infarction and received anticoagulation therapy. Osteoporosis was shown in one patient who received ADT for 41 months and was treated with bisphosphonate.

## Discussion

Our findings suggest that initial PSA, lower nadir PSA value and long-term A-ADT could be used as predictive tools for patients with high or very high risk prostate cancer treated with RT. Several randomized studies have shown that RT with long-term A-ADT could improve

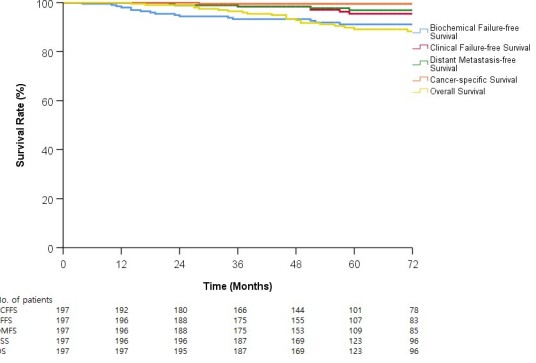

**Fig 1. Biochemical failure-frees survival, clinical failure-free survival, distant metastasis-free survival, cancer-specific survival, and overall survival in entire patients.**

**Table 2. Univariate analysis of clinical and treatment factors for treatment outcomes.**

| Variables | | No. | 5-y BCFFS | p-value | 5-y CFFS | p-value | 5-y DMFS | p-value | 5-y CSS | p-value | 5-y OS | p-value |
|---|---|---|---|---|---|---|---|---|---|---|---|---|
| Age | ≤70 years | 88 | 84.2% | 0.001 | 91.9% | 0.012 | 94.7% | 0.057 | 98.9% | 0.270 | 91.7% | 0.194 |
| | >70 years | 109 | 97.2% | | 98.9% | | 98.9% | | 100% | | 86.8% | |
| Initial PSA | ≤20 ng/mL | 98 | 95.3% | 0.014 | 97.4% | 0.051 | 98.9% | 0.055 | 98.9% | 0.310 | 87.5% | 0.273 |
| | >20 ng/mL | 99 | 87.0% | | 93.5% | | 94.8% | | 100% | | 90.4% | |
| Clinical T stage | <3b | 103 | 94.3% | 0.05 | 98.4% | 0.017 | 100% | 0.010 | 100% | 0.300 | 89.9% | 0.774 |
| | ≥3b | 94 | 87.6% | | 92.5% | | 93.8% | | 98.9% | | 88.5% | |
| Gleason Group | <5 | 152 | 92.3% | 0.171 | 96.5% | 0.077 | 97.5% | 0.110 | 100% | 0.060 | 89.4% | 0.834 |
| | ≥5 | 42 | 86.2% | | 91.3% | | 94.5% | | 97.6% | | 87.4% | |
| NCCN risk group | High | 86 | 93.3% | 0.207 | 98.1% | 0.049 | 100% | 0.026 | 100% | 0.380 | 90.4% | 0.892 |
| | Very high | 111 | 89.4% | | 93.5% | | 94.6% | | 99.1% | | 88.3% | |
| N-ADT | No | 119 | 88.2% | 0.080 | 94.2% | 0.164 | 96.3% | 0.327 | 99.1% | 0.420 | 88.1% | 0.930 |
| | Yes | 78 | 96.2% | | 97.0% | | 97.0% | | 100% | | 89.8% | |
| A-ADT | No | 37 | 74.9% | < 0.001 | 94.1% | 0.509 | 94.1% | 0.150 | 100% | 0.638 | 72.2% | < 0.001 |
| | Yes | 160 | 94.9% | | 95.7% | | 97.5% | | 99.4% | | 93.4% | |
| RT volume | Prostate | 26 | 100% | 0.091 | 100% | 0.226 | 100% | 0.314 | 100% | 0.693 | 100% | 0.200 |
| | Whole pelvis | 171 | 89.7% | | 94.7% | | 96.4% | | 99.4% | | 87.4% | |
| EQD$_{2/2}$ | ≤76.23 Gy | 111 | 91.1% | 0.584 | 94.6% | 0.523 | 96.8% | 0.140 | 100% | 0.250 | 90.9% | 0.101 |
| | >76.23 Gy | 86 | 91.8% | | 97.5% | | 97.5% | | 98.8% | | 86.3% | |
| Nadir PSA | ≤0.001 ng/mL | 99 | 99.0% | < 0.001 | 100% | 0.008 | 100% | 0.051 | 100% | 0.305 | 94.6% | 0.004 |
| | >0.001 ng/mL | 98 | 82.8% | | 90.5% | | 93.6% | | 98.9% | | 83.5% | |

BCFFS, biochemical failure-free survival; CFFS, clinical failure-free survival; DMFS, distant metastasis-free survival; CSS, cancer-specific survival; OS, overall survival; PSA, prostate-specific antigen; NCCN, National Comprehensive Care Network; N-ADT, neoadjuvant androgen deprivation therapy before radiotherapy; A-ADT, adjuvant androgen deprivation therapy after radiotherapy; RT, radiotherapy; Gy, gray; EQD$_{2/2}$, equivalent dose in 2 Gy fractions at α/β ratio of 2.0.

the survival of patients with advanced prostate cancer. In a randomized trial comparing RT alone with combination therapy with RT and long-term A-ADT for 3 years in 415 patients with locally advanced prostate cancer, 5-year CFFS (40% vs. 74%, $p = 0.0001$) and OS (62% vs. 78%, $p = 0.0002$) were significantly higher in the combination therapy group [8]. A randomized trial was conducted to evaluate the efficacy of long-term A-ADT for 977 patients with tumors extending over the prostate or regional lymph node involvement [9]. The benefit of long-term A-ADT was found in absolute 10-year CSS (78% vs. 84%, $p = 0.0052$) and OS (39% vs. 49%, $p = 0.002$). In addition, long-term A-ADT could improve survival by preventing BCF in patients with locally advanced prostate cancer [13]. The addition of long-term A-ADT reduced the risk of PSA progression by 59%, CSS by 44%, and OS by 35% compared to the patients treated with definitive RT alone. The prolonged duration of ADT could increase the survival outcomes for patients with high-risk prostate cancer [14]. In a randomized study that compared 4 months with 32 months of A-ADT in 970 patients with locally advanced prostate cancer, long-term A-ADT conferred improved 5-year CSS (96.8% vs. 95.3%, $p = 0.002$), respectively [7]. Although the addition of long-term A-ADT to RT is recommended for patients at high risk based on findings from multiple randomized trials, there is no consensus on the optimal duration of A-ADT [7–9]. In our study, A-ADT for at least 1 year resulted in a good prognosis for BCFFS and OS. The patients with A-ADT for 2 years or longer had better BCFFS than those for less than 2 years or those without A-ADT.

This study showed that patients who achieved nadir PSA levels ≤0.001 ng/mL after RT had better BCFFS and CFFS. Nadir PSA level after RT could be an independent factor for the prognosis of prostate cancer patients. Currently, there is no consensus on the absolute value of

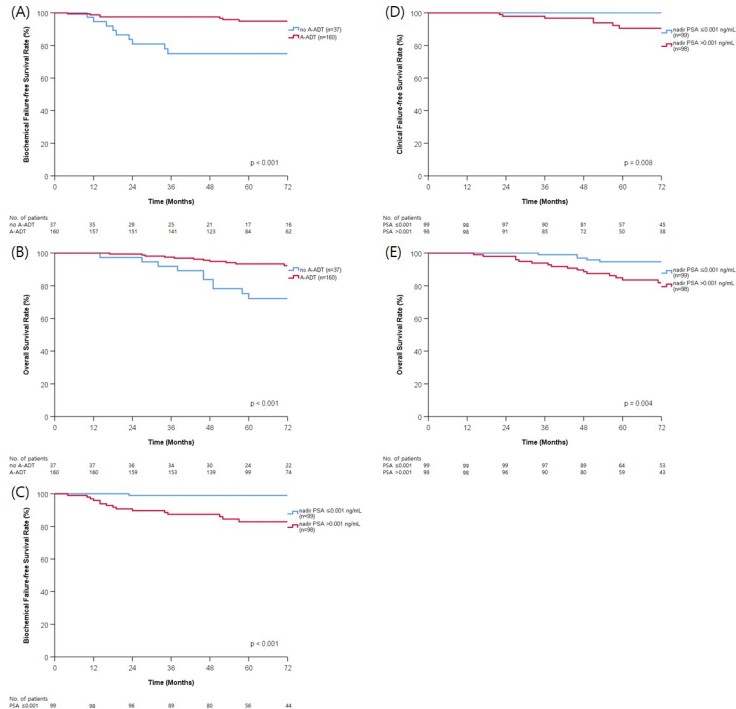

**Fig 2.** Biochemical failure-frees survival (A) and overall survival (B) according to long-term adjuvant androgen deprivation therapy (A-ADT). Biochemical failure-free survival (C), clinical failure-free survival (D), and overall survival (E) according to nadir prostate specific antigen (PSA) after radiotherapy.

nadir PSA after RT. In patients with localized prostate cancer treated with RT alone, nadir PSA <0.7 ng/mL was associated with favorable treatment outcomes [15]. Addition of adjuvant ADT for 6 months was more likely to make nadir PSA <0.5 ng/mL than RT alone, which is statistically significant in CSS from two randomized trials ($p = 0.0016$ and $< 0.0001$, respectively) [16]. Prolonging the duration of A-ADT could potentially lower the nadir PSA value. A-ADT for more than 12 months after RT was associated with nadir PSA <0.2 ng/mL significantly more frequently than ADT for less than 12 months in high-risk patients (88% in A-ADT ≥12 months vs. 74% in A-ADT <12 months, $p < 0.0001$) [17]. It has been reported

**Table 3. Multivariate analysis of clinical and treatment factors for treatment outcomes.**

| Variables | | BCFFS | | CFFS | | DMFS | | CSS | | OS | |
|---|---|---|---|---|---|---|---|---|---|---|---|
| | | HR (95% CI) | *p*-value | | *p*-value | | *p*-value | | *p*-value | | *p*-value |
| Age | ≤70 years | Reference | 0.005 | Reference | 0.026 | Reference | NS | Reference | NS | Reference | NS |
| | >70 years | 0.166 (0.047–0.581) | | 0.096 (0.012–0.760) | | | | | | | |
| Initial PSA | ≤20 ng/mL | Reference | 0.044 | Reference | NS | Reference | NS | Reference | NS | Reference | NS |
| | >20 ng/mL | 3.175 (1.030–9.788) | | | | | | | | | |
| A-ADT | No | Reference | 0.027 | Reference | NS | Reference | NS | Reference | NS | Reference | <0.001 |
| | Yes | 0.337 (0.129–0.882) | | | | | | | | 0.213 (0.094–0.484) | |
| Nadir PSA | ≤0.001 ng/mL | Reference | 0.006 | Reference | 0.021 | Reference | NS | Reference | NS | Reference | NS |
| | >0.001 ng/mL | 8.308 (1.815–38.027) | | 11.408 (1.444–90.140) | | | | | | | |

PSA, prostate-specific antigen; A-ADT, adjuvant androgen deprivation therapy after radiotherapy.

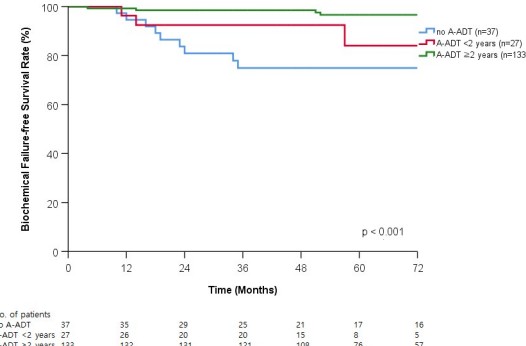

**Fig 3. Biochemical failure-frees survival curve according to duration of adjuvant androgen deprivation therapy (A-ADT).**

that a nadir PSA level of <0.06 ng/mL is a powerful prognostic factor for 5-year BCFFS for intermediate- and high-risk patients treated with RT and long-term A-ADT (96% vs. 74%, $p < 0.001$) [18]. Similarly, a nadir PSA value of <0.001 ng/mL affected 5-year BCFFS (99% vs. 82.8%, $p < 0.001$) in the current study. This nadir PSA value is comparable to the nadir PSA in patients who underwent radical prostatectomy and had a good prognosis for BCFFS [19]. This is probably due to the relatively longer duration of A-ADT with a median of 36 months in the present study compared to the median of 12 months in other studies.

The initial serum PSA level was demonstrated to predict BCFFS in prostate cancer patients treated with RT in this study. Initial serum PSA level ≤15 ng/mL had a better 3-year BCFFS of 86% compared to that of >15 ng/mL of 38% in patients with localized prostate cancer treated with RT alone [20]. In a study of 76 locally advanced prostate cancer patients treated with RT, BCFFS was 39% for those with an initial PSA value ≤15 ng/mL and 0% for those with PSA values >15 ng/mL [21]. Another study of 164 prostate cancer patients treated with RT with or without N-ADT, 2-year BCF was 11.1% for those with an PSA value ≤10 ng/mL, 23.1% for 10–19.9 ng/mL, 29.6% for 20–99.9 ng/mL, and 75% for ≥100 ng/mL [22]. In the current study, an initial PSA value of >20 ng/mL was a poor prognostic factor for BCFFS. Among 99 patients with initial PSA >20 ng/mL, patients with A-ADT + nadir PSA ≤0.001 ng/mL had better prognosis in 5-year-BCFFS (100%) than patients with no A-ADT + nadir PSA ≤0.001 ng/mL (66.7%), or with A-ADT + nadir PSA >0.001 ng/mL (83.3%), or with no A-ADT + nadir PSA >0.001 ng/mL (52.7%) ($p < 0.001$).

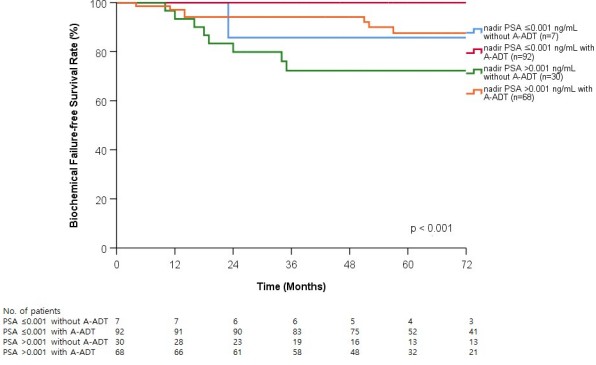

**Fig 4. Biochemical failure-frees survival curve according to long-term adjuvant androgen deprivation therapy (A-ADT) and nadir prostate specific antigen (PSA) after radiotherapy.**

**Table 4. Treatment-related acute complications.**

| Complication | Number of patients (%) | | | |
|---|---|---|---|---|
| | Grade 0 | Grade 1 | Grade 2 | Grade 3 |
| Genitourinary, acute | | | | |
| Urinary frequency | 109 (55.3) | 62 (31.5) | 26 (13.2) | 0 (0.0) |
| Urinary urgency | 193 (98.0) | 1 (0.5) | 3 (1.5) | 0 (0.0) |
| Dysuria | 172 (87.3) | 17 (8.6) | 8 (4.1) | 0 (0.0) |
| Hematuria | 194 (98.5) | 2 (1.0) | 1 (0.5) | 0 (0.0) |
| Cystitis | 180 (91.3) | 8 (4.1) | 9 (4.6) | 0 (0.0) |
| Gastrointestinal, acute | | | | |
| Anal pain | 173 (87.8) | 11 (5.6) | 13 (6.6) | 0 (0.0) |
| Diarrhea | 158 (80.2) | 26 (13.2) | 13 (6.6) | 0 (0.0) |
| Fecal incontinence | 196 (99.5) | 1 (0.5) | 0 (0.0) | 0 (0.0) |
| Proctitis | 178 (90.3) | 12 (6.1) | 7 (3.6) | 0 (0.0) |
| Rectal hemorrhage | 190 (96.4) | 5 (2.5) | 2 (1.0) | 0 (0.0) |
| Genitourinary, late | | | | |
| Urinary frequency | 153 (77.6) | 9 (4.6) | 35 (17.8) | 0 (0.0) |
| Urinary urgency | 185 (93.9) | 1 (0.5) | 11 (5.6) | 0 (0.0) |
| Dysuria | 174 (88.3) | 9 (4.6) | 14 (7.1) | 0 (0.0) |
| Hematuria | 182 (92.3) | 7 (3.6) | 7 (3.6) | 1 (0.5) |
| Cystitis | 138 (70.1) | 19 (9.6) | 40 (20.3) | 0 (0.0) |
| Urinary incontinence | 193 (98.0) | 0 (0.0) | 4 (2.0) | 0 (0.0) |
| Urinary retention | 196 (99.5) | 0 (0.0) | 0 (0.0) | 1 (0.5) |
| Gastrointestinal, late | | | | |
| Anal pain | 196 (99.5) | 0 (0.0) | 1 (0.5) | 0 (0.0) |
| Diarrhea | 194 (98.5) | 3 (1.5) | 0 (0.0) | 0 (0.0) |
| Fecal incontinence | 196 (99.5) | 1 (0.5) | 0 (0.0) | 0 (0.0) |
| Proctitis | 191 (97.0) | 3 (1.5) | 0 (0.0) | 3 (1.5) |
| Rectal hemorrhage | 188 (95.5) | 3 (1.5) | 3 (1.5) | 3 (1.5) |

It is unclear the impact of age on prognosis in prostate cancer patients treated with RT. A retrospective study reported that patients aged ≤70 years showed lower 6-year BCFFS after RT than > 70 years (82.1% vs. 94.2%, $p$ = 0.03) [23]. Patients aged ≤60 years experienced a higher rate of 5-year BCF than those aged >60 years after RT (45% vs. 35%, $p$ = 0.017) [24]. However, another study did not find a significant difference in the rate of BCF between men aged ≤60 years and >60 years after RT [25]. The effect of A-ADT on prostate cancer could be relatively weaker in older patients because of delayed recovery of testosterone [26]. A study reported that there was no difference of BCFFS between RT alone and RT with A-ADT in patients aged >70 years (94.4% VS. 94.2%, p = 0.878), while a significant difference existed in patients aged ≤70 years (82.1% VS. 94.0%, p = 0.030) [23]. In our study, the differences of BCFFS according to whether A-ADT or not was more prominent in patients aged ≤70 years (90.7% vs. 54.2%, p = 0.001) than in >70 years (98.9% vs. 90.5%. p = 0.037). Furthermore, there were more very high-risk group in patients ≤70 years than in >70 years (68.5% vs. 45.9%, p<0.001). These might be the reason that age ≤70 years was a poor prognostic factor for BCFFS in present study.

Despite the potential benefits of ADT in patients with locally advanced prostate cancer, there is a possibility of complications which negatively affect quality of life. Vasomotor hot flushes, sexual side effects, osteopenia or osteoporosis, and cardiovascular events have been

frequently reported in association with ADT [27]. Among these, cardiovascular event is life-threatening complication. There is an positive association between ADT administration and the risk of cardiovascular disease [28]. The risk of ADT-induced cardiovascular events including thromboembolism, myocardial infarction, and ischemic heart disease increases in patients with preexisting cardiovascular disease like hypertension [29]. In present study, six out of seven patients with cardiovascular events had preexisting hypertension. Patients who did not receive A-ADT had medical comorbidities more frequently than patients who received A-ADT. Among 37 patients who did not receive A-ADT, all 12 dead patients did not die of prostate cancer. However, among 11 dead patients out of 160 patients who received A-ADT, two patients died of prostate cancer and nine patients died of other causes. We thought that was why the use of A-ADT did not show significant difference in CSS but showed significant difference in OS in this study. In patients with underlying comorbidities, ADT administration should be carefully considered in view of complications.

There are several limitations to the current study. The main limitations are the retrospective design and inherent selection bias related to the heterogeneity of patients. We included and analyzed both high- and very high-risk groups in this study. The two risk groups showed significant differences in CFFS and DMFS in the univariate analysis. However, multivariate analysis did not show a significant difference in survival. Another limitation is that there is no data available for PSA immediately before RT with completion of N-ADT, which is a known prognostic factor [30]. We could not obtain accurate PSA values just before RT in almost all patients. Therefore, if there is data on PSA just before RT, it might also be a prognostic factor related to the therapeutic outcomes. Further study is needed for evaluating any significance of PSA level after N-ADT and just before RT or in combination of nadir PSA level after RT. Nevertheless, nadir PSA level after RT was found to be a significant prognostic factor for treatment outcomes in this study.

## Conclusion

In conclusion, combination therapy with RT and A-ADT could be an effective treatment in patients with locally advanced prostate cancer. For high- or very high-risk patients without comorbidities, long-term A-ADT for at least 1 year and achieving nadir PSA of ≤0.001 ng/mL could prevent biochemical failure and improve survival, consequently.

## Author Contributions

**Conceptualization:** Taek-Keun Nam.

**Data curation:** Jae-Uk Jeong, Taek-Keun Nam, Ju-Young Song, Mee Sun Yoon, Sung-Ja Ahn, Woong-Ki Chung, Ick Joon Cho, Yong-Hyub Kim, Shin Haeng Cho, Seung Il Jung, Taek Won Kang, Dong Deuk Kwon.

**Formal analysis:** Jae-Uk Jeong, Taek-Keun Nam, Ju-Young Song, Mee Sun Yoon, Sung-Ja Ahn, Woong-Ki Chung, Ick Joon Cho, Shin Haeng Cho.

**Funding acquisition:** Taek-Keun Nam.

**Investigation:** Jae-Uk Jeong, Taek-Keun Nam, Ju-Young Song, Mee Sun Yoon, Sung-Ja Ahn, Dong Deuk Kwon.

**Methodology:** Jae-Uk Jeong, Taek-Keun Nam, Ju-Young Song, Mee Sun Yoon, Sung-Ja Ahn, Woong-Ki Chung, Ick Joon Cho, Yong-Hyub Kim.

**Project administration:** Taek-Keun Nam.

**Resources:** Taek-Keun Nam.

**Supervision:** Taek-Keun Nam.

**Validation:** Jae-Uk Jeong, Taek-Keun Nam, Ju-Young Song, Mee Sun Yoon, Sung-Ja Ahn, Woong-Ki Chung, Ick Joon Cho, Yong-Hyub Kim, Shin Haeng Cho, Seung Il Jung, Taek Won Kang, Dong Deuk Kwon.

**Writing – original draft:** Jae-Uk Jeong, Taek-Keun Nam.

**Writing – review & editing:** Taek-Keun Nam, Ju-Young Song.

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
