## [Decision Letter · Decision Letter 0]

21 Jul 2020

PONE-D-20-09406

Favorable Prognosis of Patients Who Underwent Adjuvant Androgen Deprivation Therapy after Radical Radiotherapy Achieving Undetectable Levels of Prostate-Specific Antigen in High- or Very High-Risk Prostate Cancer

PLOS ONE

Dear Dr. Nam,

Thank you for submitting your manuscript to PLOS ONE. After careful consideration, we feel that it has merit but does not fully meet PLOS ONE’s publication criteria as it currently stands. Therefore, we invite you to submit a revised version of the manuscript that addresses the points raised during the review process.

We look forward to receiving your revised manuscript.

Kind regards,

Jason Chia-Hsun Hsieh, M.D. Ph.D

Academic Editor

PLOS ONE

Journal Requirements:

2. In the ethics statement in the manuscript and in the online submission form, please provide additional information about the patient records used in your retrospective study. Specifically, please ensure that you have discussed whether all data were fully anonymized before you accessed them and/or whether the IRB or ethics committee waived the requirement for informed consent. If patients provided informed written consent to have data from their medical records used in research, please include this information.

3. Thank you for including your competing interests statement; "no"

Additional Editor Comments (if provided):

Many issues existed in the current version of the manuscript. Please carefully address the points raised by the valuable reviewers.

Reviewers' comments:

Reviewer's Responses to Questions

**Comments to the Author**

1. Is the manuscript technically sound, and do the data support the conclusions?

Reviewer #1: Yes

Reviewer #2: Yes

Reviewer #3: Yes

Reviewer #4: Partly

Reviewer #5: Partly

2. Has the statistical analysis been performed appropriately and rigorously? 

Reviewer #1: No

Reviewer #2: Yes

Reviewer #3: Yes

Reviewer #4: Yes

Reviewer #5: No

3. Have the authors made all data underlying the findings in their manuscript fully available?

Reviewer #1: Yes

Reviewer #2: Yes

Reviewer #3: Yes

Reviewer #4: Yes

Reviewer #5: Yes

4. Is the manuscript presented in an intelligible fashion and written in standard English?

Reviewer #1: Yes

Reviewer #2: Yes

Reviewer #3: Yes

Reviewer #4: No

Reviewer #5: Yes

5. Review Comments to the Author

Reviewer #1: This study investigated the association treatment outcomes and related predictive factors in 197 patients with high- or very high-risk prostate cancer who underwent radical radiation therapy, using retrospective study.

Based on this study, the authors concluded that over one-year adjuvant ADT and achieving a nadir PSA≦0.001 after radiation therapy led to good treatment outcomes.

The following considerations and clarifications can improve the manuscript.

#1

Page 2 row 17: The use of long-term A-ADT significantly affected nadir PSA…

In the abstract, the authors mentioned that long-term A-ADT significantly affected nadir PSA. However, it seems that they haven’t shown this result in the manuscript.

Furthermore, if there was a significant association between these two variables, the authors should consider the multicollinearity that might affect the cox proportional hazard model.

#2

Methods section

This study is a retrospective study, so the author should explain in the method section that this study was a retrospective study.

#3

In this cohort, some patients didn’t have A-ADT.

Are there any possible reasons why some patients didn’t have A-ADT in spite of their high-risk cancer?

#4

Page 8 row 7: The most common Gleason grade…

Is this Gleason grade “Primary Gleason grade”?

#5

Page 8 row 10: The total duration of A-ADT ranged from…

Page 8 row 20: The median follow-up duration was…

It is better these results are also included in Table 1.

#6

Page 9 row 12: The results of the multivariate analysis of prognostic…

Table 3

Page 13 row 17: This might be because there were more very high-risk in patients≦70 years…

The author should explain how they chose these factors as variables in this multivariate analysis. Why didn’t they include NCCN risk groups (High or Very high) instead of age? Because of the significant association between age and NCCN risk groups that were mentioned in Page 13 row 18?

#7

Table 1, 2, 3, Fig.2

The authors used “A-ADT after RT (no vs. yes)” in Table 1, 2, 3 and manuscript.

On the other hand, they used “A-ADT (<12 months or ≧12 months)” in Fig.2.

They mentioned that A-ADT was defined as more than one-year adjuvant ADT.

Probably, “A-ADT (<12 months or ≧12 months)” is same as “A-ADT after RT (no vs. yes)”.

These statements may confuse the readers.

#8

Page 13 row 8: than patients with no A-ADT+ nadir PSA≦0.001 ng/ml,…

Please show %, like others.

#9

Page 14 row 9: However, multivariable analysis did not show a significant difference in survival.

The authors didn’t use NCCN risk group as variables in multivariable analysis.

#10

Table 3

Did authors check the overall model equation usefulness in the multivariable analysis?

#11

Fig.1 2 3 4

Please describe the number of target patients at each time below ”Time”.

#12

Fig.3 4

Please explain the statistical analysis method by which the authors compared between multiple groups.

Reviewer #2: Ad Introduction: "Preferred treatment of these patients" -> reference?

Ad Results: section: patient with 131.8 months of ADT, and you only have max 96 months follow up? So 3 years of neoadjuvant ADT treatment?

Ad discussion and results:

Having a better BCFFS with longer use of ADT, is not a conslusion, is it almost a given because of the longer use of ADT that the PSA levels stay lower, and the return to normal testosteron after the use of 2 years is longer then for 1 year, so for that analysis I would leave BBCFS out.

Did you also found something on the timing of PSA levels becoming < 0.001 ng/mL as did the group of Kuban and Crook?

Age and BCFFS -> return to normal testosteron levels after ADT use in older patients is delayed, can that be the reason for you finding?

Last conclusion on continueing ADT if there are detectable levels op PSA after RT is not true, you just given proof that it is an indepentent factor, but not that you can turn this around by given longer ADT -> so you cannot conclude this based on your data.

Reviewer #3: This is a well conducted study, there are some comments:

1: the term used in Title and manuscript: " Radical Radiotherapy" , what does " Radical " mean? usually, only radiotherapy is used

2: please describe more about the definition of " neoadjuvant ADT". " concurrent ADT". " adjuvant ADT"

the author defined neoadjuvant ADT for more than one month before RT, does the definition well accepted ?

also, adjuvant ADT for more than one year after RT? please explain the rationale and how literature support?

3: please describe more clear about ADT: the authors showed:ADT consists of a luteinizing

hormone-releasing hormone analogue (LHRH) agonist and/or an antiandrogen.

 however, only anti-androgen did not consider as standard way of ADT, if so, please excluded these cases

also, please provide details information of how these cases receive ADT, how many with LHRH-agonist, ..how many with LHRH-agonist+ anti-androgen... etc

4: about the radiotherapy, the authors showed 3D-CRT and IMRT were used, however, as we known, more modern radiotherapy used Image-guided RT ( IGRT), please describe more about this issue

5: does any cases receive brachytherapy?

6: the NCCN guidelines for prostate cancer 2020, provided many information about RT+ADT for high risk prostate cancer patients, what's new information does this study provided?

Reviewer #4: 1.It seems not a novel ideal for study on the adjuvant ADT after radical radiotherapy for high- or very high-risk prostate cancer patients, what the uniqueness exists in your study ?

2.In your study, “A-ADT for at least 1 year resulted in a good prognosis for BCFFS, CSS, and OS. Patients, who underwent A-ADT for 2 years or longer had better BCFFS than those receiving A-ADT for less than 2 years or those without A-ADT”, It seems to be the only meaningful result for discussing.

3.The 3rd paragraph of Discussion “It is unclear…………. ≤70 years than >70 years (68.5% vs. 45.9%, p<0.001)” is ambiguous and hard to be understood, please rewrite this paragraph.

4.In the line 6 of “Patients” subsection, the clinical staging of “A total of 197 patients with high- or very high-risk” shall be clearly verified.

Reviewer #5: 1.I would suggest that you may consult epidemiologist or statistician for data analyze and model selection

2.Since 62.9% patients received neoadjuvant, hormone therapy. Is it suitable to use the date of radiation therapy as zero time during analysis? Besides, you may consider to add neoadjuvant, concurrent and adjuvant time as hormone therapy periods.

3.Since there is 18 and 10 events of biochemical failure and clinical failure. It seems too few to draw conclusion.

4.Radiation therapy with 1.5-3yrs hormone therapy is the recommendation therapy in NCCN guideline. What is the novelty in your analyze.

6. PLOS authors have the option to publish the peer review history of their article (what does this mean?). If published, this will include your full peer review and any attached files.

Reviewer #1: No

Reviewer #2: No

Reviewer #3: No

Reviewer #4: No

Reviewer #5: No

---

## [Author Response · Author response to Decision Letter 0]

9 Sep 2020

Reviewer #1: This study investigated the association treatment outcomes and related predictive factors in 197 patients with high- or very high-risk prostate cancer who underwent radical radiation therapy, using retrospective study.

Based on this study, the authors concluded that over one-year adjuvant ADT and achieving a nadir PSA≦0.001 after radiation therapy led to good treatment outcomes.

The following considerations and clarifications can improve the manuscript.

#1

Page 2 row 17: The use of long-term A-ADT significantly affected nadir PSA…

In the abstract, the authors mentioned that long-term A-ADT significantly affected nadir PSA. However, it seems that they haven’t shown this result in the manuscript.

Furthermore, if there was a significant association between these two variables, the authors should consider the multicollinearity that might affect the cox proportional hazard model.

Author response: Thank you for your comments. As you pointed out, we newly inserted the description of relation between A-ADT and nadir PSA in Results section as follows; “Of the 37 patients who did not receive A-ADT, only 7 patients (7.1%) achieved nadir PSA ≤0.001 ng/mL. However, 92 patients (57.5%) of 160 patients who received A-ADT reached nadir PSA ≤0.001 ng/mL (p < 0.001)”. 

We consulted a statistician for any influence of multicollinearity of variables in multivariate analysis. We received the answer saying there was not significant multicollinearity among all 10 variables including A-ADT and nadir PSA. The VIF (Variance Inflation Factor) is known as an indicator of multicollinearity and the values of VIF were all less than 5 in 10 variables in this study, which means there was not significant multicollinearity in cox proportional hazard model (VIF of age: 1.18, initial PSA: 1.09, clinical T stage: 3.84, Gleason grade group: 1.27, NCCN risk group: 4.02, N-ADT: 1.23, A-ADT: 1.14, RT volume: 1.08, EQD2/2: 1.33, and nadir PSA: 1.16).

#2

Methods section

This study is a retrospective study, so the author should explain in the method section that this study was a retrospective study.

Author response: We described the “retrospective study” in Patients subsection in Methods.

#3

In this cohort, some patients didn’t have A-ADT.

Are there any possible reasons why some patients didn’t have A-ADT in spite of their high-risk cancer?

Author response: Four patients did not receive any type of ADT due to comorbidity of cardiovascular event. We described the reason in Results section.

#4

Page 8 row 7: The most common Gleason grade…

Is this Gleason grade “Primary Gleason grade”?

Author response: We categorized the patients in Gleason Group as used in current NCCN guideline, not by the Primary pattern. We cited the literature [12] regarding this grouping in Statistical analysis subsection of Materials and Methods. 

#5

Page 8 row 10: The total duration of A-ADT ranged from…

Page 8 row 20: The median follow-up duration was…

It is better these results are also included in Table 1.

Author response: As your comment, we included these information in Table 1 as well.

#6

Page 9 row 12: The results of the multivariate analysis of prognostic…

Table 3

Page 13 row 17: This might be because there were more very high-risk in patients≦70 years…

The author should explain how they chose these factors as variables in this multivariate analysis. Why didn’t they include NCCN risk groups (High or Very high) instead of age? Because of the significant association between age and NCCN risk groups that were mentioned in Page 13 row 18?

Author response: Thank you for your comments. We performed a multivariate analysis with all 10 variables regardless of any significance after an univariate analysis. And we found that there was not significant multicollinearity among all 10 variables via VIF (Variance Inflation Factor) analysis, aka an indicator of multicollinearity. Table 3 shows the variables having any significance in five survivals after multivariate analysis. The variables without any significance were omitted at Table 3. For more clear description, we revised the sentence regarding this point at Statistical analysis subsection of Materials and Methods as follows; “All 10 variables were used in univariate and also multivariate analysis”. 

#7

Table 1, 2, 3, Fig.2

The authors used “A-ADT after RT (no vs. yes)” in Table 1, 2, 3 and manuscript.

On the other hand, they used “A-ADT (<12 months or ≧12 months)” in Fig.2.

They mentioned that A-ADT was defined as more than one-year adjuvant ADT.

Probably, “A-ADT (<12 months or ≧12 months)” is same as “A-ADT after RT (no vs. yes)”.

These statements may confuse the readers.

Author response: As your comments, “no vs. yes” means “<12 months or ≧12 months”. So we changed the legend in Fig. 2.

#8

Page 13 row 8: than patients with no A-ADT+ nadir PSA≦0.001 ng/ml,…

Please show %, like others.

Author response: We inserted the percent in the subgroup patients like others. 

#9

Page 14 row 9: However, multivariable analysis did not show a significant difference in survival.

The authors didn’t use NCCN risk group as variables in multivariable analysis.

Author response: We used all variables including NCCN risk group (high vs. very high) used in univariate analysis. But there was no significant difference according to NCCN risk group in multivariate analysis.

#10

Table 3

Did authors check the overall model equation usefulness in the multivariable analysis?

Author response: As your comment, we checked the overall model equation usefulness in the multivariate analysis. Firstly, we tested a proportional-hazards assumption in five survival models and all satisfied the assumptions (p > 0.05). Secondly, we analyzed the Harrell's C concordance index and all five survival models had the C-indexes ranging from 0.7679 to 0.9441, which means the closer to one, the more discriminable of the model equation. 

#11

Fig.1 2 3 4

Please describe the number of target patients at each time below ”Time”.

Author response: In all figures, the number of target patients was demonstrated below “Time” as your comments.

#12

Fig.3 4

Please explain the statistical analysis method by which the authors compared between multiple groups.

Author response: The log-rank test was used in all survival comparisons among multiple groups including those figures. We described the method in Statistical analysis subsection of Materials and Methods. 

Reviewer #2: Ad Introduction: "Preferred treatment of these patients" -> reference?

Author response: We revised the sentence as follows; “Radiation therapy (RT) for these patients could be a definitive treatment among various modalities [4]”.

Ad Results: section: patient with 131.8 months of ADT, and you only have max 96 months follow up? So 3 years of neoadjuvant ADT treatment?

Author response: In this study, the actual max followed-up period and max A-ADT duration was both 131.8 months. Because of a wide range of follow-up period (14.0 ~ 131.8 months), and for more appropriate comparisons among various subgroups, we analyzed the results with patients censored at 96 months with longer follow-up. Therefore, we revised both max follow-up period and max ADT duration as 96.0 months at Patients and treatment characteristics subsection in Results and Table 1. 

Ad discussion and results:

Having a better BCFFS with longer use of ADT, is not a conclusion, is it almost a given because of the longer use of ADT that the PSA levels stay lower, and the return to normal testosterone after the use of 2 years is longer then for 1 year, so for that analysis I would leave BBCFS out.

Author response: As your comments, it’s true the longer ADT and the lower PSA level. However, some patients, although they did not receive A-ADT, could achieve nadir PSA ≤0.001 ng/mL but the patients who received A-ADT reached nadir PSA ≤0.001 ng/mL more frequently (p <0.001) as described in Prognostic factors related to treatment outcomes of Results. We tried to find out any potential treatment factors affecting treatment outcomes such as A-ADT, RT dose, and RT volume, etc. The A-ADT was the only treatment factor affecting BCFFS, CSS, and OS. So we showed the significance of A-ADT for these survival endpoints as in Table or Figure.

Did you also found something on the timing of PSA levels becoming < 0.001 ng/mL as did the group of Kuban and Crook?

Author response: We didn’t find any significance of the timing of PSA <0.001 ng/mL on treatment outcomes such as loco-regional failure or distant metastasis by either one year or two year criteria. We found that the whether the achievement of PSA <0.001 ng/mL or not was important for clinical failure as in Table 2.

Age and BCFFS -> return to normal testosterone levels after ADT use in older patients is delayed, can that be the reason for you finding?

Author response: Thank you for your detailed comments. As you pointed out, the biochemical failure could be relatively weaker in older patients because of delayed recovery of testosterone. In our study, the differences of BCFFS according to the receiving A-ADT or not was more prominent in patients aged ≤70 years (90.7% vs. 54.2%, p = 0.001) than in >70 years (98.9% vs. 90.5%. p = 0.037). We rewrote the paragraph about age and ADT affecting BCFFS in Discussion section.

Last conclusion on continuing ADT if there are detectable levels op PSA after RT is not true, you just given proof that it is an independent factor, but not that you can turn this around by given longer ADT -> so you cannot conclude this based on your data.

Author response: As your recommendation, the conclusion was changed as follows. “In conclusion, combination therapy with RT and A-ADT could be an effective treatment in patients with locally advanced prostate cancer. For high- or very high-risk patients without comorbidities, long-term A-ADT for at least 1 year and achieving nadir PSA of ≤0.001 ng/mL could prevent biochemical failure and improve survival, consequently. 

Reviewer #3: This is a well conducted study, there are some comments:

1: the term used in Title and manuscript: " Radical Radiotherapy" , what does " Radical " mean? usually, only radiotherapy is used

Author response: As your recommendation, we removed the term of ‘Radical’.

2: please describe more about the definition of " neoadjuvant ADT". " concurrent ADT". " adjuvant ADT"

Author response: We described the definition of ADT in Treatments subsection of Materials and Methods.

the author defined neoadjuvant ADT for more than one month before RT, does the definition well accepted ?

Author response: We changed the definition of neoadjuvant ADT duration to for more than 2 months according to the literature (J Clin Oncol 2008;26(15):2497-504). Fortunately, all patients who received N-ADT received for at least 2 months in our study. 

also, adjuvant ADT for more than one year after RT? please explain the rationale and how literature support?

Author response: As you pointed out, adjuvant ADT for 1.5-3 years is currently recommended for high risk patients. In our study, BCSSF was better in the group of ADT for more than 2 years than in the group of ADT less than 2 years (figure 3). However, some literatures showed that ADT for more than 1 year could decrease nadir PSA effectively and improve biochemical failure-free survival (Am J Clin Oncol;40(4):348-52, Radiat Oncol 2017;12(1):149). One randomized trial of dose escalation study using EBRT or brachytherapy for high risk patients, adjuvant ADT was performed for 1 year (Int J Radiat Oncol Biol Phys 2017;98(2):275-85). So, we used the definition of A-ADT duration as for equal or more than 1 year in this study.

3: please describe more clear about ADT: the authors showed:ADT consists of a luteinizing

hormone-releasing hormone analogue (LHRH) agonist and/or an antiandrogen.

 however, only anti-androgen did not consider as standard way of ADT, if so, please excluded these cases

also, please provide details information of how these cases receive ADT, how many with LHRH-agonist, ..how many with LHRH-agonist+ anti-androgen... etc

Author response: All patients received ADT consisted of LHRH-agonist with or without an antiandrogen. We described detailed information about the number of patients received LHRH-agonist + anti-androgen or LHRH-agonist alone in Table 1.

4: about the radiotherapy, the authors showed 3D-CRT and IMRT were used, however, as we known, more modern radiotherapy used Image-guided RT ( IGRT), please describe more about this issue

Author response: We described about using IGRT technique in Methods section. “To verify the patient’s position, daily electronic portal images and weekly cone beam CT images were obtained using a linear accelerator, or daily megavoltage CT was performed using helical Tomotherapy.”

5: does any cases receive brachytherapy?

Author response: No patients received brachytherapy. We added the description in Treatments subsection of Materials and Methods.

6: the NCCN guidelines for prostate cancer 2020, provided many information about RT+ADT for high risk prostate cancer patients, what's new information does this study provided?

Author response: We think the novelty of this study is the level of nadir PSA after RT (≤0.001 ng/mL) is much lower than other studies. This is probably due to the relatively longer duration of A-ADT with a median of 36 months in the present study compared to the median of 12 months in other studies. Nadir PSA value ≤0.001 ng/mL is also a good prognostic factor in patients received radical prostatectomy in some references. To our best knowledge, this is the first study that the nadir PSA value of 0.001 ng/mL is an independent factor on treatment outcomes in patients treated with RT and ADT in high and very high risk group.

Reviewer #4: 

1. It seems not a novel ideal for study on the adjuvant ADT after radical radiotherapy for high- or very high-risk prostate cancer patients, what the uniqueness exists in your study ?

Author response: We think the novelty of this study is the level of nadir PSA after RT (≤0.001 ng/mL) is much lower than other studies. This is probably due to the relatively longer duration of A-ADT with a median of 36 months in the present study compared to the median of 12 months in other studies. Nadir PSA value ≤0.001 ng/mL is also a good prognostic factor in patients received radical prostatectomy in some references. To our best knowledge, this is the first study that the nadir PSA value of 0.001 ng/mL is an independent factor on treatment outcomes in patients treated with RT and ADT in high and very high risk group.

2. In your study, “A-ADT for at least 1 year resulted in a good prognosis for BCFFS, CSS, and OS. Patients, who underwent A-ADT for 2 years or longer had better BCFFS than those receiving A-ADT for less than 2 years or those without A-ADT”, It seems to be the only meaningful result for discussing.

Author response: Thank you for your comments.

3. The 3rd paragraph of Discussion “It is unclear…………. ≤70 years than >70 years (68.5% vs. 45.9%, p<0.001)” is ambiguous and hard to be understood, please rewrite this paragraph.

Author response: Thank you for your detailed comments. The biochemical failure could be reduced in older patients because of delayed recovery of testosterone. In our study, the differences of BCFFS according to whether A-ADT or not was more prominent in patients aged ≤70 years (90.7% vs. 54.2%, p = 0.001) than in >70 years (98.9% vs. 90.5%. p = 0.037). We rewrote the paragraph about age and ADT affecting BCFFS in Discussion section.

4. In the line 6 of “Patients” subsection, the clinical staging of “A total of 197 patients with high- or very high-risk” shall be clearly verified.

Author response: We described the breakdown of clinical staging in entire patients at Table 1.

Reviewer #5: 

1. I would suggest that you may consult epidemiologist or statistician for data analyze and model selection.

Author response: We consulted the statistician for the statistical appropriateness in this study, such as multicollinearity among variables via variance inflation factors or model equation usefulness analysis via Harrell’s C-index in multivariate analysis. The statistician said all results were appropriately analyzed and all models selected were useful. 

2. Since 62.9% patients received neoadjuvant, hormone therapy. Is it suitable to use the date of radiation therapy as zero time during analysis? Besides, you may consider to add neoadjuvant, concurrent and adjuvant time as hormone therapy periods.

Author response: As you point out, N-ADT could reduce the volume of prostate cancer and lower the pre-RT PSA level. Lower pre-RT PSA level after N-ADT is known to show better radiotherapy outcomes. However, whether N-ADT or not was not significant in all survival endpoints in this study. Therefore, we tried to figure out the effect of adding A-ADT for high and very risk patients and analyzed from the start date of RT as zero time. We consider a further study for figure out any prognostic significance affecting outcomes according to pre-RT PSA level after N-ADT.

3. Since there is 18 and 10 events of biochemical failure and clinical failure. It seems too few to draw conclusion.

Author response: It seems that few events was probably due to the relatively longer duration of A-ADT with a median of 36 months in the present study. However, most events on biochemical failure (16 of 18) and clinical failure (9 of 10) occurred in the patients with nadir PSA > 0.001 ng/mL.

4. Radiation therapy with 1.5-3yrs hormone therapy is the recommendation therapy in NCCN guideline. What is the novelty in your analyze.

Author response: We think the novelty of this study is the level of nadir PSA after RT (0.001 or less ng/mL) is much lower than other studies. This is probably due to the relatively longer duration of A-ADT with a median of 36 months in the present study compared to the median of 12 months in other studies. Nadir PSA value ≤0.001 ng/mL is also a good prognostic factor in patients received radical prostatectomy in some references. To our best knowledge, this is the first study that the nadir PSA value of 0.001 ng/mL is an independent factor on treatment outcomes in patients treated with RT and ADT in high and very high risk group.

Thank you all for your efforts of thorough review and invaluable comments.

---

## [Decision Letter · Decision Letter 1]

6 Nov 2020

PONE-D-20-09406R1

Favorable Prognosis of Patients Who Underwent Adjuvant Androgen Deprivation Therapy after Radiotherapy Achieving Undetectable Levels of Prostate-Specific Antigen in High- or Very High-Risk Prostate Cancer

PLOS ONE

Dear Dr. Nam,

Thank you for submitting your manuscript to PLOS ONE. After careful consideration, we feel that it has merit but does not fully meet PLOS ONE’s publication criteria as it currently stands. Therefore, we invite you to submit a revised version of the manuscript that addresses the points raised during the review process.

We look forward to receiving your revised manuscript.

Kind regards,

Jason Chia-Hsun Hsieh, M.D. Ph.D

Academic Editor

PLOS ONE

Additional Editor Comments (if provided):

Reviewer #3 found some data discrepancy, which requires further clarification.

Reviewers' comments:

Reviewer's Responses to Questions

**Comments to the Author**

1. If the authors have adequately addressed your comments raised in a previous round of review and you feel that this manuscript is now acceptable for publication, you may indicate that here to bypass the “Comments to the Author” section, enter your conflict of interest statement in the “Confidential to Editor” section, and submit your "Accept" recommendation.

Reviewer #1: All comments have been addressed

Reviewer #3: (No Response)

Reviewer #5: All comments have been addressed

2. Is the manuscript technically sound, and do the data support the conclusions?

Reviewer #1: Yes

Reviewer #3: Partly

Reviewer #5: Partly

3. Has the statistical analysis been performed appropriately and rigorously? 

Reviewer #1: Yes

Reviewer #3: No

Reviewer #5: N/A

4. Have the authors made all data underlying the findings in their manuscript fully available?

Reviewer #1: Yes

Reviewer #3: Yes

Reviewer #5: Yes

5. Is the manuscript presented in an intelligible fashion and written in standard English?

Reviewer #1: Yes

Reviewer #3: Yes

Reviewer #5: Yes

6. Review Comments to the Author

Reviewer #1: (No Response)

Reviewer #3: The major concerns is that this study did not provide novel information regarding the treatment of prostate cancer.

Reviewer #5: 1.

In your table1, the percentage of 5-yr CFFS, 5-yr DMFS, 5-yr CSS in no A-ADT group is 94.1%, 94.1%,89%. Why the percentage of 5-yr CSS is lower than 5-yr CFFS, 5-yr DMFS. Does it mean that some patients died due to prostate cancer without disease progression or metastasis? You may give the numerator and denominator in each column. In multiple variable analysis, the HR is 0.111 in ADT group compared with non- ADT group, this data is strange.

.2.

The androgen duration in your cohort is 12-96 months. It means some patients were still under androgen therapy during 5 yrs followed-up. Hence it is not surprisingly that these patient were not experienced biochemical failure. Hence more important is DMFS, CSS in your cohort.

3.

Besides, which is more important, A-ADT, or androgen duration (N-ADT+A-ADT)?

If the zero time is date of androgen usage, were there any difference in your results?

4. Since radiation therapy with 1.5-3yrs hormone therapy is the recommendation therapy in NCCN guideline. Hence, less than 1.5 yrs hormone therapy is inadequate therapy. You may exclude these patients to show the neediness of longer androgen duration.

5. Since the novelty of your study is that psa<0.001 is an important prognostic factor. You may show more analysis regarding psa<0.001.

7. PLOS authors have the option to publish the peer review history of their article (what does this mean?). If published, this will include your full peer review and any attached files.

Reviewer #1: No

Reviewer #3: No

Reviewer #5: No

---

## [Author Response · Author response to Decision Letter 1]

14 Nov 2020

Reviewer #1: 

(No Response)

Reviewer #3: 

The major concerns is that this study did not provide novel information regarding the treatment of prostate cancer.

Reviewer #5: 

1. In your table1, the percentage of 5-yr CFFS, 5-yr DMFS, 5-yr CSS in no A-ADT group is 94.1%, 94.1%,89%. Why the percentage of 5-yr CSS is lower than 5-yr CFFS, 5-yr DMFS. Does it mean that some patients died due to prostate cancer without disease progression or metastasis? You may give the numerator and denominator in each column.

Author response: We calculated the CSS with the information of cause of death from Department of Cancer Statistics at our institute and also from Ministry of Health in our country. However, according to your comments, we found out that some information of cause of death were wrongly gathered as ICD of prostate cancer (C61), actually even without prostate cancer progression. So we reviewed the hospital records of dead patients again and found their last follow-up date at hospital very close to date of death. We confirmed no evidence of prostate cancer progression in 5 out of 6 patients, who were initially regarded as an event of CSS. We revised the 5 cases of death as a censored at date of death without prostate cancer progression and accordingly recalculated CSS. We revised the CSS in manuscript and Table 2 and 3. The significance of A-ADT was lost in view of CSS in uni- and multivariate analysis. 

In multiple variable analysis, the HR is 0.111 in ADT group compared with non- ADT group, this data is strange.

Author response: It means that the HR for CSS in A-ADT patients was significantly lower than those of no A-ADT by 0.111 times at initial manuscript. However, after we recalculated CSS with updated information of cause of death, the significance of A-ADT was lost in view of CSS in uni- and multivariate analysis. 

2. The androgen duration in your cohort is 12-96 months. It means some patients were still under androgen therapy during 5 yrs followed-up. Hence it is not surprisingly that these patient were not experienced biochemical failure. Hence more important is DMFS, CSS in your cohort.

Author response: In multivariate analysis, we found that patients with A-ADT for at least 12 months survived longer significantly in view of BCFFS or OS. However, after we recalculated CSS with updated information of cause of death, the significance of A-ADT was lost in view of CSS in uni- and multivariate analysis. 

3. Besides, which is more important, A-ADT, or androgen duration (N-ADT+A-ADT)?

If the zero time is date of androgen usage, were there any difference in your results?

Author response: N-ADT (for ≥2months prior to RT) was performed in 78 (39.6%) patients but we didn’t find any significant difference between N-ADT and no N-ADT groups in all treatment outcomes in this study. Therefore, we tried to figure out the effect of adding A-ADT with the day after the completion of RT as of zero time. 

4. Since radiation therapy with 1.5-3yrs hormone therapy is the recommendation therapy in NCCN guideline. Hence, less than 1.5 yrs hormone therapy is inadequate therapy. You may exclude these patients to show the neediness of longer androgen duration.

Author response: As you pointed out, adjuvant ADT for 1.5-3 years is currently recommended for high risk patients. Also in our study, the patients with ≥18 months developed BCF much less frequently than the patients with less than 18 months or no A-ADT (5.8% vs. 25.6%). Moreover, BCSSF was better in the group of ADT for more than 2 years than less than 2 years (Figure 3). In this study, six patients received A-ADT for 12 - 18 months and of them, one patient developed BCF at 9 months after RT even during A-ADT. There were concerns about ADT complications which could negatively affect the quality of life with longer duration of A-ADT. Some patients discussed with urologists an intermittent A-ADT or interruption of ADT after one year of A-ADT for their quality of life or due to concurrent medical illness. Some literatures showed that ADT for more than 1 year could decrease nadir PSA effectively and improve biochemical failure-free survival (Am J Clin Oncol;40(4):348-52, Radiat Oncol 2017;12(1):149). One randomized trial of dose escalation study using EBRT or brachytherapy for high risk patients, adjuvant ADT was performed for 1 year (Int J Radiat Oncol Biol Phys 2017;98(2):275-85). So, for various reasons, we used the definition of A-ADT duration as for at least 1 year to figure out the effect of A-ADT in this study.

5. Since the novelty of your study is that psa<0.001 is an important prognostic factor. You may show more analysis regarding psa<0.001.

Author response: The detailed analysis of nadir PSA ≤0.001 ng/mL after RT was presented in a Table 2 and 3. We think the novelty of this study is the level of nadir PSA after RT (0.001 or less ng/mL) is much lower than other studies. This is probably due to the relatively longer duration of A-ADT with a median of 36 months in the present study compared to other studies. Nadir PSA value ≤0.001 ng/mL is also a good prognostic factor in patients received radical prostatectomy in some references. To our best knowledge, this is the first study that the nadir PSA value of 0.001 ng/mL is an independent factor on treatment outcomes in patients treated with RT and ADT in high and very high risk group.

We appreciate all your comment

---

## [Decision Letter · Decision Letter 2]

20 Jan 2021

PONE-D-20-09406R2

Favorable Prognosis of Patients Who Received Adjuvant Androgen Deprivation Therapy after Radiotherapy Achieving Undetectable Levels of Prostate-Specific Antigen in High- or Very High-Risk Prostate Cancer

PLOS ONE

Dear Dr. Nam,

Thank you for submitting your manuscript to PLOS ONE. After careful consideration, we feel that it has merit but does not fully meet PLOS ONE’s publication criteria as it currently stands. Therefore, we invite you to submit a revised version of the manuscript that addresses the points raised during the review process.

ACADEMIC EDITOR: One data discrepancy still exists in the revised manuscript. Please explain that in a detailed response.

We look forward to receiving your revised manuscript.

Kind regards,

Jason Chia-Hsun Hsieh, M.D. Ph.D

Academic Editor

PLOS ONE

Additional Editor Comments (if provided):

One data discrepancy still exists in the revised manuscript. Please explain that in a detailed response.

Reviewers' comments:

Reviewer's Responses to Questions

**Comments to the Author**

1. If the authors have adequately addressed your comments raised in a previous round of review and you feel that this manuscript is now acceptable for publication, you may indicate that here to bypass the “Comments to the Author” section, enter your conflict of interest statement in the “Confidential to Editor” section, and submit your "Accept" recommendation.

Reviewer #3: All comments have been addressed

Reviewer #5: All comments have been addressed

2. Is the manuscript technically sound, and do the data support the conclusions?

Reviewer #3: Partly

Reviewer #5: Partly

3. Has the statistical analysis been performed appropriately and rigorously? 

Reviewer #3: Yes

Reviewer #5: N/A

4. Have the authors made all data underlying the findings in their manuscript fully available?

Reviewer #3: Yes

Reviewer #5: Yes

5. Is the manuscript presented in an intelligible fashion and written in standard English?

Reviewer #3: Yes

Reviewer #5: Yes

6. Review Comments to the Author

Reviewer #3: Reviewer think that the revised manuscript lacking of novel information regarding the treatment of prostate cancer.

Reviewer #5: Since long-term adjuvant hormone therapy can not prolong CSS. the finding of long-term adjuvant hormone therapy can prolong OS is a strange finding and need to be clarified.

Administration of long-term A-ADT

significantly predicted favorable BCFFS (p = 0.027) and OS (p < 0.001) in multivariate

analysis.

7. PLOS authors have the option to publish the peer review history of their article (what does this mean?). If published, this will include your full peer review and any attached files.

Reviewer #3: **Yes: **Shu-Pin Huang

Reviewer #5: No

---

## [Author Response · Author response to Decision Letter 2]

21 Jan 2021

Reviewer #5: Since long-term adjuvant hormone therapy can not prolong CSS. the finding of long-term adjuvant hormone therapy can prolong OS is a strange finding and need to be clarified.

Answer: Patients who did not receive A-ADT had medical comorbidities more frequently than patients who received A-ADT. Among 37 patients who did not receive A-ADT, all 12 dead patients did not die of prostate cancer. However, among 11 dead patients out of 160 patients who received A-ADT, two patients died of prostate cancer and nine patients died of other causes. We thought that was why the use of A-ADT did not show significant difference in CSS but showed significant difference in OS in this study.

We added this description in Discussion section. 

We appreciate your review comment.

---

## [Decision Letter · Decision Letter 3]

1 Mar 2021

Favorable Prognosis of Patients Who Received Adjuvant Androgen Deprivation Therapy after Radiotherapy Achieving Undetectable Levels of Prostate-Specific Antigen in High- or Very High-Risk Prostate Cancer

PONE-D-20-09406R3

Dear Dr. Nam,

We’re pleased to inform you that your manuscript has been judged scientifically suitable for publication and will be formally accepted for publication once it meets all outstanding technical requirements.

Kind regards,

Jason Chia-Hsun Hsieh, M.D. Ph.D

Academic Editor

PLOS ONE

Additional Editor Comments (optional):

All the questions were answered adequately.

Reviewers' comments:

Reviewer's Responses to Questions

**Comments to the Author**

1. If the authors have adequately addressed your comments raised in a previous round of review and you feel that this manuscript is now acceptable for publication, you may indicate that here to bypass the “Comments to the Author” section, enter your conflict of interest statement in the “Confidential to Editor” section, and submit your "Accept" recommendation.

Reviewer #6: All comments have been addressed

2. Is the manuscript technically sound, and do the data support the conclusions?

Reviewer #6: Yes

3. Has the statistical analysis been performed appropriately and rigorously? 

Reviewer #6: Yes

4. Have the authors made all data underlying the findings in their manuscript fully available?

Reviewer #6: Yes

5. Is the manuscript presented in an intelligible fashion and written in standard English?

Reviewer #6: Yes

6. Review Comments to the Author

Reviewer #6: The authors have addressed all comments and revised appropriately. This revised manuscript is currently acceptable for publication in this journal.

7. PLOS authors have the option to publish the peer review history of their article (what does this mean?). If published, this will include your full peer review and any attached files.

Reviewer #6: No

---

## [Editor Report · Acceptance letter]

3 Mar 2021

PONE-D-20-09406R3 

Favorable Prognosis of Patients Who Received Adjuvant Androgen Deprivation Therapy after Radiotherapy Achieving Undetectable Levels of Prostate-Specific Antigen in High- or Very High-Risk Prostate Cancer 

Dear Dr. Nam:

I'm pleased to inform you that your manuscript has been deemed suitable for publication in PLOS ONE. Congratulations! Your manuscript is now with our production department. 

Kind regards, 

on behalf of

Dr. Jason Chia-Hsun Hsieh 

Academic Editor

PLOS ONE